# A physically interpretable data-driven surrogate model for wake steering

Balthazar Arnoldus Maria Sengers[1], Matthias Zech[2], Pim Jacobs[1], Gerald Steinfeld[1], and Martin Kühn[1]

[1]ForWind, Institute of Physics, Carl von Ossietzky University Oldenburg, Küpkersweg 70, 26129 Oldenburg, Germany
[2]German Aerospace Center (DLR), Institute of Networked Energy Systems, Carl-von-Ossietzky-Str. 15, 26129 Oldenburg, Germany

**Correspondence:** Balthazar Sengers (balthazar.sengers@uni-oldenburg.de)

**Abstract.** Wake steering models for control purposes are typically based on analytical wake descriptions tuned to match experimental or numerical data. This study explores whether a data-driven surrogate model with a high degree of physical interpretation can accurately describe the redirected wake. A linear model trained with large eddy simulation data estimates wake parameters such as deficit, center location and curliness from measurable inflow and turbine variables. These wake parameters are then used to generate vertical cross sections of the wake at desired downstream locations. In a validation considering eight boundary layers ranging from neutral to stable conditions, the far wake's trajectory, curl and available power are accurately estimated. A significant improvement in accuracy is shown in a benchmark study against two analytical wake models, especially under derated operating conditions and stable atmospheric stratifications. Even though the results are not directly generalizable to all atmospheric conditions, locations or turbine types, the outcome of this study is encouraging.

## 1   Introduction

Wind turbine wakes cause considerable power losses and increased loads at downstream machines. Control strategies to mitigate these negative effects are gaining support in the wind energy community. In particular wake steering, or wake redirection through intentional yaw misalignment (Dahlberg and Medici, 2003; Wagenaar et al., 2012), is regarded as a promising control strategy. A yaw misalignment introduces a lateral thrust force component, which redirects the downstream wake and generates two counter-rotating vortices around upper and lower tip height that curl the wake into a kidney shape (Howland et al., 2016). Numerical simulations (e.g. Gebraad et al., 2016; Fleming et al., 2018; Hulsman et al., 2020), wind tunnel experiments (e.g. Campagnolo et al., 2016; Bartl et al., 2018; Bastankhah and Porté-Agel, 2019) and free field campaigns (e.g. Fleming et al., 2017, 2019, 2020; Bromm et al., 2018) have demonstrated the potential of an increased wind farm power production when utilizing the wake steering concept. The efficacy of wake steering is strongly dependent on turbine operation and atmospheric inflow characteristics, such as the turbine thrust coefficient (Jimenez et al., 2010), atmospheric stability (Vollmer et al., 2016), wind shear (Schottler et al., 2017) and turbulence intensity (Bastankhah and Porté-Agel, 2016).

Wake steering controllers regulating the turbine yaw angle are often based on simple wake models that can describe the downstream wake under different inflow conditions. These models, such as those available in the FLORIS framework (NREL, 2020), are typically based on a simplified analytical description of the momentum conservation equations for stationary inflow

conditions. When combined with a dynamic controller, wind direction variability can be included (Rott et al., 2018; Simley et al., 2020). The performance of these wake steering controllers, and therefore the accuracy of the underlying wake models, is essential for a successful application of wake steering as a control strategy in a real wind farm.

Frequently used in recent years is the wake model based on Gaussian self-similarity (Bastankhah and Porté-Agel, 2014, 2016; Abkar and Porté-Agel, 2015; Niayifar and Porté-Agel, 2016). Combining wake deficit and wake deflection models, the Gaussian (GAUS) model uses turbulence intensity as an atmospheric inflow parameter. It was validated against field measurements in Annoni et al. (2018) and used as a controller in a field campaign in Fleming et al. (2019). A disadvantage of this model is the negligence of the counter-rotating vortices generated with yaw misalignment and consequently the absence of wake curling. In addition, it does not account for the initial wake deflection caused by the torque induced wake rotation in sheared inflow (Zahle and Sørensen, 2008). The curl model (Martínez-Tossas et al., 2019) accounts for these phenomena by explicitly including vortices in a model based on linearized Reynolds-averaged Navier-Stokes equations. Having a strong physical basis, it is able to include a wide range of atmospheric conditions and allows flexibility in the wake shape generation. A disadvantage is the high computational expense compared to GAUS. For this reason, King et al. (2021) proposed to include the vortex description of the curl model into GAUS in the Gaussian-Curl Hybrid (GCH) model. This incorporates the initial wake deflection and even secondary wake steering, the deflection of the wake of a downstream turbine by the vortices generated by the yawed upstream turbine (Fleming et al., 2018), but not the curling of the wake itself. In addition, the model includes a wake recovery term representing added entrainment by the vortices generated due to yaw misalignment. Fleming et al. (2020) showed promising results when using GCH as controller input in a free field campaign.

These analytical models contain parameters that can be tuned to match numerical or experimental data. In addition, data can be used to formulate parameterized error terms (Schreiber et al., 2020). However, completely data-driven wake models remain rare and those that exist generally use complex machine learning models with a low interpretability (e.g. Göçmen and Giebel, 2018; Ti et al., 2020). This is remarkable, since simple data-driven models are proven to be able to describe complex physical phenomena (Brunton et al., 2016) and are already widely used for prediction purposes, including wind power (Stathopoulos et al., 2013; Messner and Pinson, 2019) and power curve predictions (Marčiukaitis et al., 2017). Although analytical models are presumably more robust, especially when the data set is small, the maximum achievable accuracy is also limited as it is not feasible to develop one model for all scenarios. An analytical model will not be able to capture features for which equations were not in place, hence constant updates to the model code are necessary (e.g. Abkar et al., 2018; Bastankhah et al., 2022). With the community demanding that wake models include increasingly more complex features (e.g. the wake curl), data-driven models become interesting as they can directly capture these features when enough data is available.

The objective of this study is to explore the potential of a Data-driven wAke steeRing surrogaTe model (DART) that retains a high degree of physical interpretation. It is investigated whether the curled wake can accurately be described by a set of measurable inflow and turbine variables, and how the use of these variables can be optimized in an interpretable model. Next, the potential of this surrogate model is systematically assessed by evaluating its performance with large eddy simulations (LES) for a reference wind speed under a range of atmospheric conditions and subsequently benchmarking it against two analytical

wake models (GAUS and GCH). Lastly, the surrogate model's generalizability to all atmospheric conditions, locations and turbine types is discussed.

## 2 Large eddy simulations

In this study data are generated with revision 3475 of the PArallelized Large eddy simulation Model (PALM, Maronga et al. (2020)), which uses a non-hydrostatic incompressible Boussinesq approximation of the Navier-Stokes equations and the Monin-Obukhov Similarity Theory to describe surface fluxes. In the boundary layer, the grid on a right-handed Cartesian coordinate system is regularly spaced with $\Delta = 5$ m, while above the boundary layer height the vertical grid size increases with 6 % per cell to save computational costs. The Coriolis parameter corresponds to that at 55 °N. Default numerical schemes are used, the main ones being a third-order Runge-Kutta scheme for time integration, a fifth-order Wicker-Skamarock advection scheme for the momentum equations, Deardorff's 1.5-order turbulence closure parameterization for subgrid-scale turbulence and an iterative multigrid scheme for the horizontal boundary conditions. The simulation chain consists of a precursor simulation to generate realistic inflow conditions and a subsequent main simulation that contains one turbine.

### 2.1 Precursor simulations

Inflow conditions with realistic turbulent features are generated from an initially laminar flow by adding random perturbations in a precursor simulation with cyclic horizontal boundary conditions. To study the potential of DART under different inflow conditions, eight boundary layers (BLs) ranging from a neutral to a strongly stable BL are used as reference inflow conditions, all having approximately the same wind direction and wind speed at hub height. As reported by Vollmer et al. (2016), wake steering is ineffective in a convective boundary layer, which is therefore not considered in this study. Due to the large computational expense it was not possible to increase the number of simulations. Although these eight BLs do not capture the great variability of the free field, it is considered sufficient to provide a proof of concept for data-driven models.

The total domain size and simulation time vary between the BLs and are determined empirically until convergence to a stationary state occurs and are dependent on the size of the largest eddies that explicitly need to be resolved. The details of the precursor simulations are summarized in Table 1.

BL1 and BL2 portray neutral conditions with roughness lengths representing low crops ($z_0 = 0.1$ m) and parkland ($z_0 = 0.5$ m), which are typical landscapes found in Northern Germany. Following Basu et al. (2008), constant cooling rates at the surface are prescribed to generate stable BLs, where BL3 and BL4 represent weakly stable ($\partial\theta \, \partial t^{-1} = -0.125$ K h$^{-1}$), and BL5 and BL6 strongly stable conditions ($\partial\theta \, \partial t^{-1} = -0.25$ K h$^{-1}$, following Beare and Macvean (2004)). Two additional (near) neutral BLs are generated, one representing grassland ($z_0 = 0.03$ m) and one having a very small negative sensible heat flux ($H$ = -0.001 K m s$^{-1}$), which is in the acceptable range defined in Basu et al. (2008).

Stationary inflow conditions are taken at 2.5 rotor diameters (D) upstream of the turbines simulated in the main simulations (Sect. 2.2) and averaged over a line of size 2 D in the crosswise direction and a period of 60 minutes. These inflow conditions are assumed to be undisturbed, hence far enough from the turbine that induction does not play a role. Typical inflow

**Table 1.** Summary of simulation parameters and classification into neutral (NBL), near neutral (NNBL), weakly stable (WSBL) and stable (SBL) boundary layers. The size ($L_{x,\mathrm{p}}$, $L_y$, $L_z$) of the domains is normalized by the rotor diameter (D = 126 m). All parameters are identical in precursor and main simulations, except for the domain size which is extended in the streamwise direction ($L_{x,\mathrm{m}}$). $t_\mathrm{p}$ is the simulated time of the precursor run, $u_g$ and $v_g$ the geostrophic wind, $\partial\theta\,\partial t^{-1}$ the heating rate, $H$ the sensible heat flux and $z_0$ the surface roughness length.

| | | $t_\mathrm{p}$ | $L_{x,\mathrm{p}}$ | $L_{x,\mathrm{m}}$ | $L_y$ | $L_z$ | $u_g$ | $v_g$ | $\partial\theta\,\partial t^{-1}$ | $H$ | $z_0$ |
| | | [h] | [D] | [D] | [D] | [D] | [m s$^{-1}$] | [m s$^{-1}$] | [K h$^{-1}$] | [K m s$^{-1}$] | [m] |
|---|---|---|---|---|---|---|---|---|---|---|---|
| **BL1** | NBL | 28 | 40.6 | 61.0 | 20.3 | 14.0 | 10.115 | -3.969 | - | - | 0.1 |
| **BL2** | NBL | 28 | 40.6 | 61.0 | 20.3 | 14.0 | 10.595 | -5.572 | - | - | 0.5 |
| **BL3** | WSBL | 25 | 27.9 | 50.0 | 14.0 | 8.4 | 9.952 | -5.115 | -0.125 | - | 0.1 |
| **BL4** | WSBL | 45 | 27.9 | 50.0 | 14.0 | 8.4 | 10.607 | -6.447 | -0.125 | - | 0.5 |
| **BL5** | SBL | 20 | 11.4 | 30.5 | 7.6 | 4.6 | 9.500 | -5.170 | -0.250 | - | 0.1 |
| **BL6** | SBL | 20 | 11.4 | 30.5 | 7.6 | 4.6 | 10.565 | -6.585 | -0.250 | - | 0.5 |
| **BL7** | NBL | 40 | 40.6 | 61.0 | 20.3 | 14.0 | 9.609 | -3.193 | - | - | 0.03 |
| **BL8** | NNBL | 40 | 40.6 | 61.0 | 20.3 | 14.0 | 9.831 | -3.488 | - | -0.001 | 0.1 |

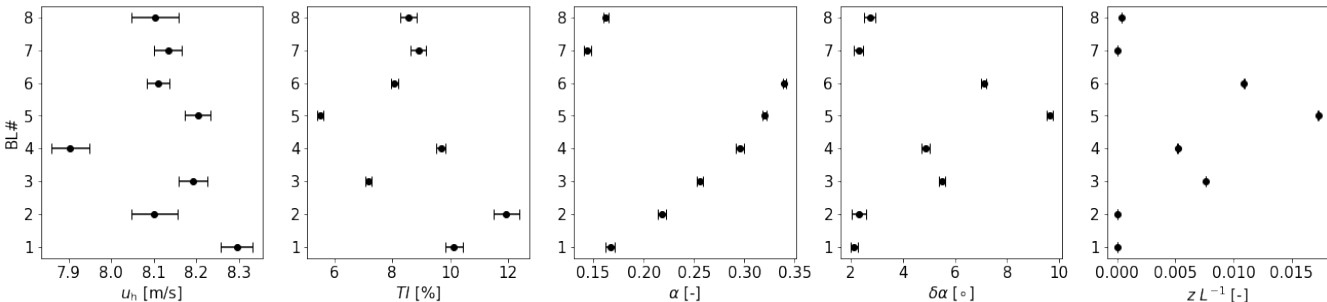

**Figure 1.** Summary of inflow parameters (60 min averages), given as mean (dots) and standard deviation (whiskers) over the 15 main simulations performed in each BL (5 yaw angles times 3 pitch angles). Considered are wind speed ($u_\mathrm{h}$) and turbulence intensity at hub height ($TI$), wind shear ($\alpha$) and veer ($\delta\alpha$) over the rotor area and the Obukhov stability parameter ($z\,L^{-1}$) at $z = 2.5$ m. Equations for these variables can be found in Table 3.

parameters are displayed in Fig. 1, indicating that the wind speed is comparable for all simulations, but that the atmospheric conditions vary. A more stable boundary layer, indicated by a larger Obukhov stability parameter ($z\,L^{-1}$), typically has a higher shear ($\alpha$) and veer ($\partial\alpha$) and lower turbulence intensity ($TI$). The spread of the parameters between the main simulations (see Sect. 2.2) in the same boundary layer, indicated by the standard deviation as whiskers in Fig. 1, is small enough to be neglected.


## 2.2 Main simulations

After generating stationary inflow conditions with a precursor, simulations with one turbine are performed. Information on turbulence characteristics from the precursor simulation is fed to the main simulation by adding a turbulent signal to a fixed mean inflow (turbulence recycling method) far upstream of the turbine. A radiation boundary condition ensures undisturbed

outflow downstream of the simulated turbine. The size of the recycling area is equal to the domain size of the precursor simulation and the domain size of the main simulation is only extended in streamwise direction by placing a turbine at $x = 6$ D downstream of the recycling area. Wake data until $x = 10$ D are used for analysis, but the domain is extended to $x = 13$ D to eliminate blockage effects. The total simulation time is 80 minutes: the first 20 minutes are considered spin-up time, the last 60 minutes are used for analysis.

The simulated turbine is an Actuator Disc Model with Rotation (ADMR) representing a 5MW NREL turbine, having a hub height of 90 m and a rotor diameter $D$ of 126 m (Jonkman et al., 2009), as implemented in Dörenkämper et al. (2015). Turbine yaw angles ($\phi$) of -30°, -15°, 0°, 15°, and 30° are simulated, where a positive yaw angle is here defined as a clockwise rotation of the turbine when looking from above. Pitch angles ($\beta$) of 0°, 2.5°, and 5° are simulated to study the effect of the thrust force on downstream wake characteristics. This adds up to a total of 120 main simulations with one turbine, i.e. 15 turbine settings

(5 yaw angles times 3 pitch angles) for each of the 8 inflow conditions. The effect of $\phi$ and $\beta$ on the thrust coefficient $C_{\mathrm{T}}$ is illustrated in Fig. 2, illustrating that the effect of the turbine yaw angle of the thrust coefficient is approximately symmetric around zero.

It should be noted that smaller step sizes for yaw and pitch angles would be preferred, as these step sizes could be too coarse when utilizing a regression based model (Sect. 3.3). This can lead to deviating estimates when interpolating to values far

away from these set points (e.g. for $\phi = 7.5°$). Increasing the step size would, however, lead to more simulations, which was

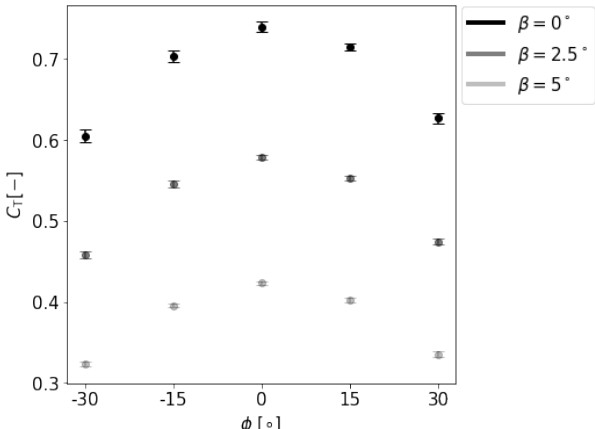

**Figure 2.** Overview of the effect of yaw angle $\phi$ and pitch angle $\beta$ on thrust coefficient $C_{\mathrm{T}}$. Whiskers indicate the standard deviation between all eight BLs.

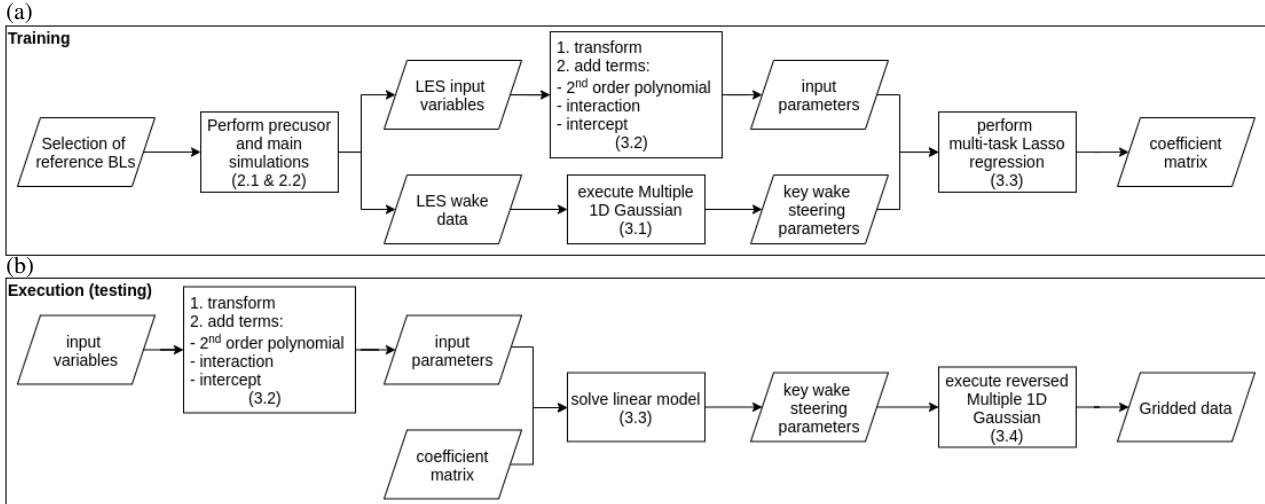

**Figure 3.** Flowchart describing the training (a) and execution (b) procedure of DART. Between parenthesis is indicated in what section the process is described. The coefficient matrix generated in (a) is used in (b).

computationally not feasible and not deemed necessary to show proof of concept.

The wake is described using the normalized wake deficit, defined as $u_{nd} = \frac{u_{wake} - u_\infty}{u_{\infty,\mathrm{h}}}$, where $u_{wake}$ represents the observed wind speed in the wake, $u_\infty$ the undisturbed inflow 2.5 D upstream at the same height and $u_{\infty,\mathrm{h}}$ the undisturbed inflow at hub height. It is assumed that the advection velocity is constant in streamwise direction (assumption of frozen turbulence) and that

the wake behaves as a passive tracer in the ambient wind (Larsen et al., 2008).

## 3 Development of the Data-driven wAke steeRing surrogaTe model

This section describes the development of the Data-driven wAke steeRing surrogaTe model (DART). It should be noted that many different kinds of data-driven models exist. For the purpose of this exploratory study, the focus was to develop a simple regression model that performs well on small data sets without the risk of overfitting.

Figure 3 displays a flowchart of the training and execution (including testing) procedure. The respective sections in which each step is explained, is indicated between parenthesis. DART is trained with the LES data representing reference inflow conditions (BLs) described in Sect. 2. From the wake data, key wake steering variables are deducted by executing the Multiple 1D Gaussian method explained in Sect. 3.1. Additionally, input variables are extracted at 2.5 D upstream, and several operations are performed (Sect. 3.2) to determine the final input parameters. A multi-task Lasso regression (Sect. 3.3) is subsequently

performed to generate a coefficient matrix.

This matrix can be used in the execution (testing) of the model to estimate the key wake steering parameters for new inflow conditions. The same operations as in the training procedure are done on the input variables to obtain the input parameters, after which the linear model (Sect. 3.3) is solved to estimate the key wake steering parameters. A reversed version of the

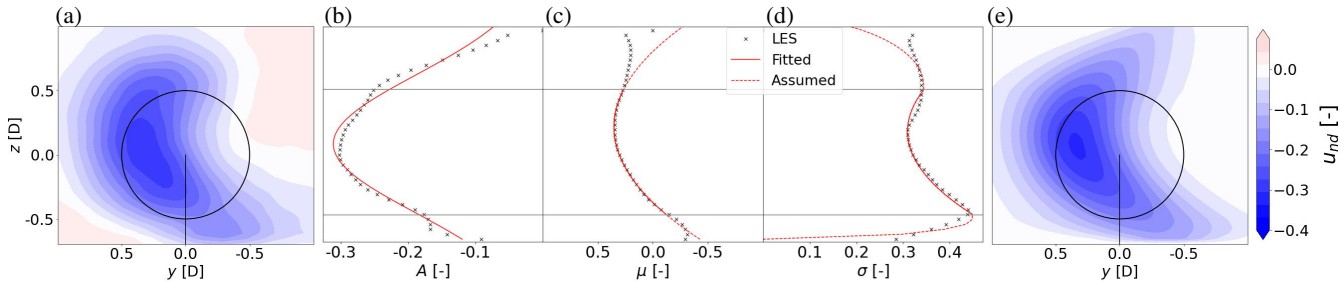

**Figure 4.** Exemplary figures (BL1, $\phi = +30°$, $x = 5$ D) illustrating the key wake steering parameters. (a) Normalized wake deficit cross section (contour) of original LES data. (b) The local normalized wake center deficits $\boldsymbol{A}$, (c) local wake center positions $\boldsymbol{\mu}$, (d) local wake widths $\boldsymbol{\sigma}$. Black crosses indicate LES, red solid lines the relation fitted in according to the Multiple 1D Gaussian method (Sect. 3.1) and red dashed lines the assumed continuation in the reversed Multiple 1D Gaussian composition method (Sect. 3.4). e) Cross section (contour) of the normalized wake deficit after applying the reversed Multiple 1D Gaussian composition method.

Multiple 1D Gaussian model can then be executed (Sect. 3.4) to obtain gridded wake data. During model development, this
wake estimation can be compared to the original LES data. One can experiment with different input variables and operations to determine what set of input parameters gives the most accurate solution (Sect. 3.5). This last step has not been included in Fig. 3 to reduce clutter.

### 3.1 Defining key wake steering parameters

A data-driven model will not be able to produce a full multidimensional flow field, but rather estimate parameters describing
the wake at desired downstream positions. Since curled wakes are considered, key wake steering parameters are in this study retrieved with the Multiple 1D Gaussian method (Sengers et al., 2020). In the example below, the wake of a turbine with a +30° yaw angle in BL1 at $x = 5$ D is considered (Fig. 4a). This method fits a simple 1D Gaussian at every vertical level ($k = 1...K$) where information is available to obtain a set of local normalized wake center deficits ($\boldsymbol{A} = A_1...A_K$), wake center positions ($\boldsymbol{\mu} = \mu_1...\mu_K$) and wake widths ($\boldsymbol{\sigma} = \sigma_1...\sigma_K$). Subsequently, another Gaussian can be fitted through the local wake center
deficits in the vertical (Fig. 4b) to find the overall normalized wake center deficit ($A_z$) and vertical position with respect to hub height ($\mu_z$), as well as the vertical extension of the wake ($\sigma_z$). The local wake center position and width at vertical level $k$ that corresponds to $\mu_z$ are subsequently considered as lateral wake center position ($\mu_y$) relative to the turbine location and wake width ($\sigma_y$). Next, by fitting a second order polynomial through the local wake center positions between upper and lower tip height (Fig. 4c), one obtains a measure for the *curl* (coefficient of quadratic term) and *tilt* (coefficient of linear term) of the
wake. An expression for the wake width as a function of height is found by repeating this step for the local wake widths (Fig. 4d) to obtain coefficients $s_a$ and $s_b$. After this procedure, the wake can be described by the set of dimensionless parameters displayed in Table 2.

Note that this method cannot accurately capture the splitting of the wake in two separate cells, which might occur under strong

**Table 2.** Defined dimensionless key wake steering parameters. The normalized wake deficit is computed as described in Sect. 2.2. All length parameters are nondimensionalized by the rotor diameter D.

| Scalar Parameter | Symbol |
| --- | --- |
| Amplitude normalized wake deficit | $A_z$ |
| Lateral wake center displacement | $\mu_y$ |
| Vertical wake center displacement | $\mu_z$ |
| Width wake center height | $\sigma_y$ |
| Vertical extend | $\sigma_z$ |
| Curl | $curl$ |
| Tilt | $tilt$ |
| Quadratic wake width parameter | $s_a$ |
| Linear wake width parameter | $s_b$ |

veer as discussed in Vollmer et al. (2016). Such cases will result in inaccurate values for the key wake steering parameters and
should be filtered out before applying the regression model described in Sect. 3.3.

## 3.2 Input parameters

A regression model (Sect. 3.3) is used to estimate the key wake steering parameters in Table 2. A set of measurable inflow and turbine variables is used as input parameters, which are made dimensionless to make the model more universally applicable, at least within the variability found between the simulations in this study. This set of parameters is presented in Table 3.
Although these input parameters might all have their own isolated effect on the wake propagation, they are heavily correlated in LES as shown in Fig. 5. One can identify several highly correlated input clusters, representing 1) yaw [$\phi$], 2) atmospheric inflow [$\delta\alpha$, $\alpha$, $z\,L^{-1}$, $TI$] and 3) turbine variables [$C_\mathrm{T}$, $C_\mathrm{Q}$, $TSR$]. Note that wind speed is not included, since it is approximately constant in all simulations and correlated with both inflow and turbine parameters. A high correlation between variables indicates that they contain much of the same information. Providing the same information to a model multiple times is futile
as it will not improve the accuracy. For this reason, it is hypothesized that reasonable accuracy can be achieved with the regression model as long as one variable from each input cluster is included. This would reduce the number of model parameters and would give the user freedom to choose parameters based on preference and availability. However, since the input variables are not perfectly correlated, the information they contain is slightly different and including both variables can increase the model's accuracy. For this reason, two versions of the surrogate model having a different number of variables are experimented
with, see Sect. 3.5. Although a high correlation between input variables is usually undesirable in regression problems due to multicollinearity, Sect. 3.3 explains that this is not an issue due to the regression model used in this study.

This regression model is linear, so to include nonlinear relations the input variables can be transformed using reciprocal ($f(x) = x^{-1}$), exponential ($f(x) = e^x$), logarithmic ($f(x) = \ln(x)$) or square root ($f(x) = \sqrt{x}$)) transformations. All these

**Table 3.** Set of dimensionless input parameters. $dir$ is the wind direction [°], $z$ is the height above the surface [m] $\overline{u_{\mathrm{h}}}$ and $\sigma_{u_{\mathrm{h}}}$ are the mean and standard deviation of the wind speed at hub height [m s$^{-1}$], $u_{eff}$ is rotor effective wind speed [m s$^{-1}$], $T$ is thrust [N], $Q$ is torque [N m] and $\omega$ is rotor speed [rad s$^{-1}$]. Subscript $_{\mathrm{ut}}$ indicates upper tip and $_{\mathrm{lt}}$ lower tip height.

| Variable | Symbol | Calculated |
|---|---|---|
| Turbine yaw angle | $\phi$ | $\phi$ |
| Veer | $\delta\alpha$ | $dir_{\mathrm{ut}} - dir_{\mathrm{lt}}$ |
| Shear | $\alpha$ | $ln\frac{u_{\mathrm{ut}}}{u_{\mathrm{lt}}} / ln\frac{z_{\mathrm{ut}}}{z_{\mathrm{lt}}}$ |
| Obukhov stability parameter | $z\,L^{-1}$ | $2.5/L$ |
| Turbulence intensity | $TI$ | $\sigma_{u_{\mathrm{h}}}/\overline{u_{\mathrm{h}}}$ |
| Thrust coefficient | $C_{\mathrm{T}}$ | $T / (0.5\,\rho\,u_{eff}^2\,\pi\,(D/2)^2)$ |
| Torque coefficient | $C_{\mathrm{Q}}$ | $Q / (0.5\,\rho\,\pi\,u_{eff}^2\,(D/2)^3)$ |
| Tip speed ratio | $TSR$ | $\omega\,(D/2) / u_{\mathrm{h}}$ |

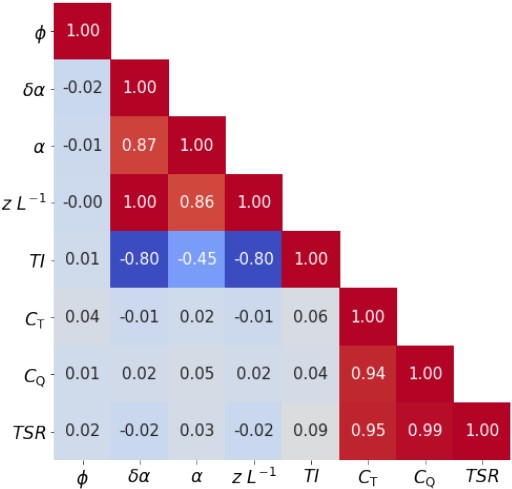

**Figure 5.** Correlation matrix of the dimensionless input parameters in LES. Colors indicate a positive (red) or negative (blue) correlation

transformations have been tested in the procedure described in Sect. 3.5. In addition to the transformed variables, second-order polynomial and interaction terms are added, as well as an intercept (unity), extending the set of input parameters.

### 3.3 Regression model

Since the LES data set has a relatively small sample size, a linear model is chosen as they perform well on small sample sizes, reduce the risk of overfitting compared to more complex Machine Learning models and are highly interpretable (Hastie et al.,

2009).

The regression is formulated as a linear model in matrix form

$$\underset{(n\times d)}{\boldsymbol{Y}} = \underset{(n\times p)}{\boldsymbol{X}} \times \underset{(p\times d)}{\boldsymbol{B}}. \tag{1}$$

which estimates the output variable $\boldsymbol{Y}$ based on the design matrix $\boldsymbol{X}$ and coefficient matrix $\boldsymbol{B}$. Matrix dimensions indicated in Eq. (1) represent the sample size $n$, number of downstream distances $d$ and number of input parameters $p$. Note that $p$ contains the transformed variables, their second-order and interaction terms, as well as intercepts. Since these parameters

are highly correlated and not all relevant, the coefficients are determined based on a Lasso regression method as introduced by Tibshirani (1996). This guarantees a shrinkage of the number of variables through a regularization parameter found by cross-validation. Relevant input parameters are isolated from irrelevant parameters by multiplying the latter with a coefficient of zero, effectively eliminating them from Eq. 1. Multicollinearity is therefore not an issue in Lasso, contrary to Ordinary Least Squares, as typically only one parameter is chosen from a set of highly correlated input parameters. This reduces the

number of input parameters, increasing the interpretability of the model. Additionally, it is desired that the same set of input parameters is used to estimate the output variable at all downstream distances. This is guaranteed in the multi-task Lasso method introduced by Obozinski et al. (2006), which is implemented in the multi-task Lasso algorithm from the *scikit-learn* Python library (Pedregosa et al., 2011). See Appendix A for further explanation.

Whereas fitting the regression coefficients is more complex than Ordinary Least Squares fitting, the estimations of the key

wake steering variables in the testing or execution phase are generated through simple matrix multiplication as shown in Eq. (1). The algorithm is therefore highly interpretable, easy to implement and computational inexpensive.

### 3.4 Wake composition: reversed Multiple 1D Gaussian

The coefficient matrix $\boldsymbol{B}$ can be used to estimate the key wake steering parameters in Table 2 from inflow variables. This information is used to compose a vertical cross section of the wake deficit using the reverse of the Multiple 1D Gaussian

method described in Sect. 3.1. The amplitude of the normalized wake deficit $\hat{\boldsymbol{A}}$ at each height ($k = 1..K$) can be computed by simply filling out the Gaussian function using $A_z, \mu_z, \sigma_z$. Similarly, local wake center positions $\hat{\boldsymbol{\mu}}$ and local wake widths $\hat{\boldsymbol{\sigma}}$ can be found by filling out a second order polynomial. Additional assumptions outside of the rotor area are that the curl continues (red dashed line in Fig. 4c) and the wake width can be described by an ellipse between lower tip and surface and between upper tip and wake top (red dashed line in Fig. 4d). Finally, a simple 1D Gaussian can be filled out at every vertical level using

the information from $\hat{\boldsymbol{A}}, \hat{\boldsymbol{\mu}}, \hat{\boldsymbol{\sigma}}$, resulting in a two-dimensional grid filled with $u_{nd}$ values (Fig. 4e). Comparing this composed wake to the original LES in Fig. 4a, one can see that this simple description still contains much of the original information. The shape of the wake is conserved, as well as the displacement of the wake center. The maximum deficit of the composed wake center appears to be slightly larger than in LES. Additionally, in the composition the maximum wake deficit is always in the center (definition of a Gaussian), which is not necessarily true in LES or reality.

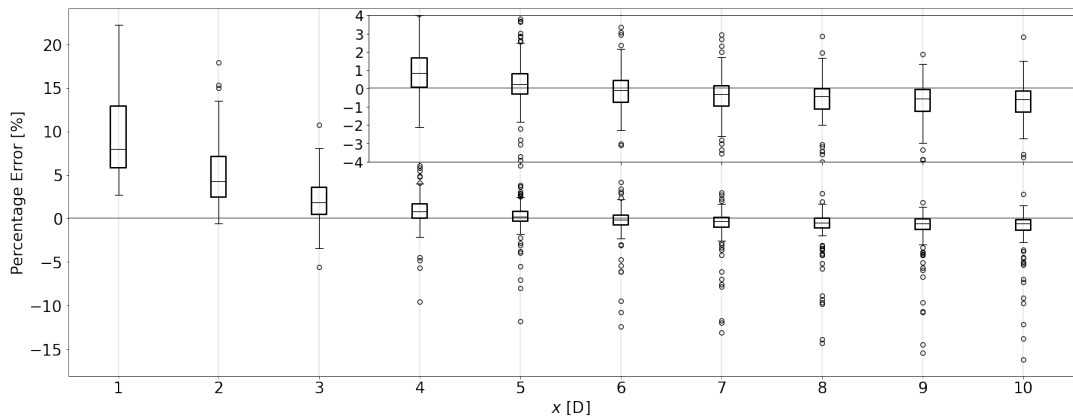

**Figure 6.** Accuracy of the wake composition procedure expressed as a percentage error of available power of a virtual downstream turbine. At each downstream distance, data from all 120 simulations are considered. The subplot in the top right zooms in to $4\,\mathrm{D} \leq x \leq 10\,\mathrm{D}$. Axes labels correspond to those of the main plot.

### 3.4.1 Wake composition validation

The procedure described in Sect. 3.4 is repeated for all 120 simulations and $1\,\mathrm{D} \leq x \leq 10\,\mathrm{D}$ at every D. The metric used here to evaluate the accuracy of this method is the percentage error of available power in the rotor area of the composed wake relative to when computed with the original LES wind field ($PE\ [\%] = (P_{\mathrm{comp}} - P_{\mathrm{LES}})/P_{\mathrm{LES}} * 100$). A few things can be noted by studying the results shown in Fig. 6. The composition shows a large systematic positive bias in the near wake ($x \leq 3$ D). This is due to the so-called double bell shape of the near wake, with a speed up region around hub height. When attempting to fit this with a simple 1D Gaussian, the deficit in the rotor area is underestimated, resulting in a positive percentage error. For this reason, the near wake will be excluded from analysis in the remainder of this work. Further downstream ($x \geq 8$ D) a small negative systematic bias can be identified, which is due to the 'top-hat' shape of the wake deficit as a result of temporal averaging. This is not captured by a Gaussian function and will on average result in an overestimation of the wake deficit amplitude. The large (negative) outliers typically indicate cases where the wake does not have a Gaussian shape, such as the separation in two cells under strong veer. The median error in the region $4\,\mathrm{D} \leq x \leq 10\,\mathrm{D}$ is, however, smaller than 1 %.

### 3.5 Feature selection

Numerous combinations of input parameters are possible. This includes choosing from the variables presented in Table 3, as well as which of the five transformations proposed in Sect. 3.2 to use. In order to find the most accurate solution, all combinations are tested. The combination that provides the minimum absolute percentage error of available power over all training data, i.e. all considered simulations and downstream distances ($4\,\mathrm{D} \leq x \leq 10\,\mathrm{D}$), is sought. When using all eight variables presented in Table 3, denoted DART-8, only the variable transformations need to be decided. The number of possible combinations is proportional to the number of transformation to the power of the number of variables. All five transformations are tested on all

variables except for $\phi$, for which the logarithmic and square root transformations have been omitted because negative values occur. This results in a total of $3^1 * 5^7 = 234375$ possible combinations. Not only is using all variables computationally expensive, as will be discussed in Sect. 4.1, operationally it is also unlikely that all variables are routinely obtained due to high costs. As hypothesized in Sect. 3.2, using one variable from each input cluster is already expected to produce accurate results. To test this, a version of DART with only three variables, denoted DART-3, is considered. Allowing each variable to be chosen and transformed, the total number of possible combinations is a multiplication of the possible combinations of each input cluster. In total, $(1 * 3) * (4 * 5) * (3 * 5) = 900$ possible combinations are tested to find the optimal set of input parameters. It should be noted that other feature selection procedures could be considered to reduce the computational expense needed for training, but enhancing the training procedure was considered outside of the scope of the current work.

## 3.6 Benchmark models

DART is benchmarked against the Gaussian (GAUS) and the Gaussian-Curl Hybrid (GCH) models present in version 2.2.2 of the FLORIS framework (NREL, 2020). Although secondary steering is not studied here, the GCH is still included because of its incorporation of initial wake deflection and the added wake recovery term. Both models share the same tuning parameters for the far wake onset ($\alpha_{\text{floris}}, \beta_{\text{floris}}$) and wake recovery rate ($k_{a,\text{floris}}, k_{b,\text{floris}}$). Analogous to the training of DART discussed in Sect. 3.5, the values of the tuning parameters is determined by minimizing the APE of available power over all considered simulations and downstream distances ($4\,\text{D} \leq x \leq 10\,\text{D}$). Information on inflow (e.g. $u_{\text{h}}, TI$) is taken from the LES data. The models are trained independently of each other and will therefore have different values for the tuning parameters.

The data used for the tuning include simulations with yaw and pitch angles. FLORIS adjusts the thrust coefficient numerically for yaw angles, but not for pitch angles. For this reason, the thrust coefficient lookup table was adjusted by the ratio $C_{\text{T,pitch}}/C_{\text{T,nopitch}}$ found in LES (Fig. 2).

## 4 Results

## 4.1 Performance on training data

This section displays the performance of the Data-driven wAke steeRing surrogaTe model (DART) and the benchmark models when using all 120 simulations for training or tuning. In Sect. 4.2 and 4.3 a validation of the model with testing data will be shown.

Following the feature selection procedure as described in Sect. 3.5, the optimal combinations of input parameters of DART-8 was found to be the set $\{\phi,\ \delta\alpha,\ \alpha^{-1},\ \ln(z\,L^{-1}),\ TI^{-1},\ C_{\text{T}}^{-1},\ \sqrt{C_{\text{Q}}},\ TSR^{-1}\}$, and for DART-3 $\{\phi,\ \delta\alpha^{-1},\ \ln(C_{\text{T}})\}$. Figure 7 compares the performance of these versions to that of the benchmark models as a function of downstream distance. The shaded areas indicate a significant improvement (green), insignificant difference (yellow) or significant decline (red) of the DART accuracy compared to the best performing benchmark model. Statistical significance is determined using an independent Welch's t-test on the absolute percentage error with a p-value < 0.05. This test assumes a normal distribution, but can deal with

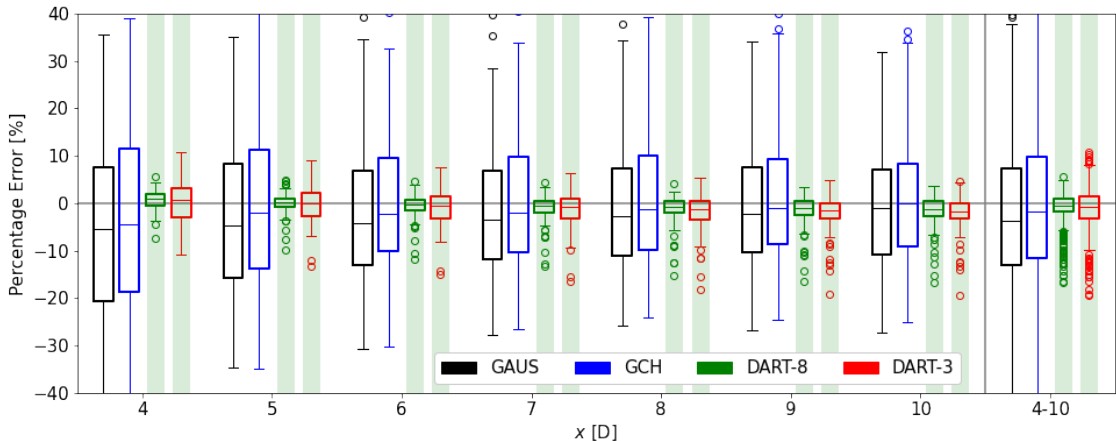

**Figure 7.** Performance of all models on training data displayed as percentage error of available power. In black GAUS, in blue GCH, in green DART-8 and in red DART-3. The boxes on the far right (labeled 4-10) include all simulations and all distances. The shaded areas indicate a significant improvement (green), insignificant difference (yellow) or significant decline (red) of the accuracy of DART compared to the benchmark models.

unequal variances between data sets.

From Fig. 7 it is clear that both DART-8 and DART-3 consistently provide significantly more accurate results than GAUS and GCH. Most striking is the variability of the benchmark models that is an order of magnitude larger than that of DART. The reason for this will be systematically evaluated in Sect. 4.2 and 4.3. The systematic error, indicated by the median, is however very similar for all models. Comparing the two benchmark models, it is clear that GCH consistently gives a higher power
estimate than GAUS due to the added wake recovery term. The accuracy of DART-8 is higher than that of DART-3, especially closer to the turbine. This is attributed to the stronger wake deficit closer to the turbine, as the wake center deficit $A_z$ exhibits a larger range of possible values closer to the turbine. For instance in the training data at $x = 4$ D, the range is -0.6 $\leq A_z \leq$ -0.21, whereas at $x = 10$ D the range is -0.27 $\leq A_z \leq$ -0.09. Estimations with the same relative error therefore bear a larger absolute error closer the turbine. Having access to more information, DART-8 consistently has a smaller relative error estimating $A_z$
than DART-3, which has a larger effect on the available power estimates closer to the turbine.

The order of magnitude of computational costs needed to train the models on a single node is displayed in Table 4. Computational expenses needed to generate the LES database are not considered. The benchmark models tune their parameters in approximately 7.5 h (GAUS) and 8.25 h (GCH). DART's training procedure is split up in different stages. The column *Iteration* refers to the regression fitting to obtain the coefficient matrix **B** (Sect. 3.3) and the calculation of the absolute percentage
error of available power at 4 D $\leq x \leq$ 10 D (Sect. 3.5). This can be carried out in seconds, in which fewer variables result in faster fitting. The column *Combinations* indicates the number of possible combinations that need to be tested (Sect. 3.5). The *Total* training time is then simply the number of combinations to be tested times the execution time of one iteration. Because

**Table 4.** Model training (DART) or tuning (GAUS and GCH) time using all 120 simulations and 7 (4 D $\leq x \leq$ 10 D) downstream distances. *Iteration* times are expressed as the mean over the first 100 iterations. DART's *Total* is a simple multiplication of *Iteration* and *Combinations*.

|          | Iteration | Combinations | Total |
|----------|-----------|--------------|-------|
|          | [s]       | [-]          | [h]   |
| **GAUS**   | -         | -            | 7.5   |
| **GCH**    | -         | -            | 8.25  |
| **DART-8** | 148       | 234375       | 9635  |
| **DART-3** | 45        | 900          | 11.25 |

of its large number of possible combinations, DART-8's total training time would be over a year on a single node, which is not operationally feasible. To generate the results in Fig. 7, the training process was heavily parallelized. With 900 possible combinations DART-3 can be trained in approximately 11.25 h, which is only slightly more than the benchmark models. As mentioned in Sect. 3.5, the training procedure could be enhanced, but this was considered outside of the scope of the current work. Even though Fig. 7 shows a small accuracy gain of DART-8 over DART-3, the computational costs to train DART-8 are much larger and measuring all these variables in the free field is impractical. For these reasons, it is decided to only consider DART-3 in the remainder of this study.

## 4.2 Performance on testing data

A simple leave-one-out cross-validation technique is used to discuss the performance of DART compared to the benchmark models. The models are trained or tuned with seven out of the eight BLs (Fig. 1) and tested on the remaining one representing a new inflow condition. Eight evaluations can therefore be performed, i.e. each BL being tested once. Note that for each evaluation a set of optimal parameters and transformations are determined, which can differ from DART-3 in Fig. 7. Similarly, GAUS and GCH are tuned again, resulting in new values for their tuning parameters. Since the models show similar behavior in relation to the downstream distance as discussed in Sect. 4.1, here only the collective result over 4 D $\leq x \leq$ 10 D is discussed. Figure 8 presents the results of this validation procedure. For all BLs, DART-3 shows a significant improvement over GAUS and GCH. The systematic biases (indicated by the medians) are similar for all models in the order of a few percent, but the variability is greatly reduced in DART-3. The main reason for this is that the benchmark models do not include a pitch angle parameter $\beta$. Although the $C_\mathrm{T}$ tables in the models are corrected in this study, the tunable parameters do not account for this. To clarify, LES finds a decreasing wake size (in both horizontal and vertical extent) with increasing $\beta$. This is accurately captured by DART-3, but GAUS and GCH produce a wake of similar size independent of $\beta$ or $C_\mathrm{T}$. The inclusion of this effect is a notable improvement of DART that is important for control strategies such as axial induction control (e.g. Corten and Schaak, 2003; van der Hoek et al., 2019).

Furthermore, BL5 contains the worst results for all models. Figure 1 indicates that this is an extreme case as it has the highest Obukhov stability parameter and veer along with the lowest turbulence intensity. This is problematic for the models, since it is an inflow condition unlike anything it was trained for. This indicates a limited generalizability of all models and caution is

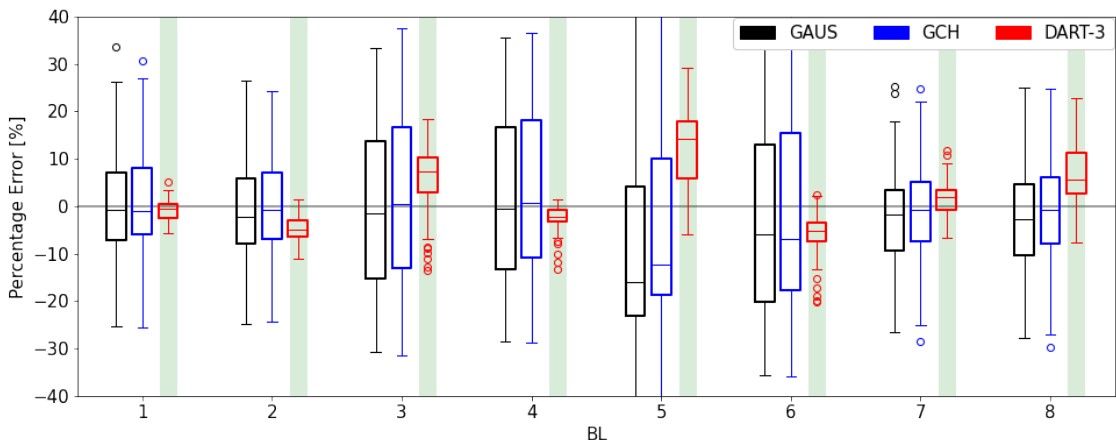

**Figure 8.** Performance of GAUS (black), GCH (blue) and DART-3 (red) using a leave-one-out cross-validation technique. Performance is displayed as a percentage error of available power. Each box includes data from 15 main simulations and $4\,\mathrm{D} \leq x \leq 10\,\mathrm{D}$. The shaded areas again indicate a significant improvement (green), insignificant difference (yellow) or significant decline (red) of the accuracy of DART-3 compared to the benchmark models.

needed when applying them in conditions that differ greatly from those used for training. This will be further discussed in Sect. 5.1.

### 4.3 Operation without derating

For a fair comparison between DART-3 and the benchmark models, this section only considers simulations representing operation without derating the turbine ($\beta = 0°$). The training (selection of parameters for DART-3) and tuning (tuning parameters of GAUS and GCH) has been repeated and the results of the leave-one-out cross-validation technique are displayed in Fig. 9. The variability of the benchmark models in (near) neutral conditions (BL 1, 2, 7 and 8) decreases considerably, but DART-3 still produces significantly more accurate results. In (weakly) stable boundary layers (BLs 3 to 6) GAUS and GCH still show a large variability and occasionally a large systematic bias, which is not true for DART-3. These results suggest that DART-3 outperforms the benchmark models especially under stable stratifications, those conditions where wake steering is deemed most effective.

Furthermore, the model performance is assessed for partial wake operation. Figure 10 compares the models when the downstream turbine is moved 0.5 D to the left (from the upstream observer's point of view). Generally, the variability is greatly reduced since the deficit is smaller. The benchmark models display a systematic negative bias in all BLs, which is not true for DART-3. Only in BL8 DART-3 does not show a significant improvement over the benchmark models, but no satisfying explanation has been found why exactly this BL displays this behavior.

A case study is displayed in Fig. 11a that presents the LES wake in a weakly stable boundary layer (BL3) for a turbine with

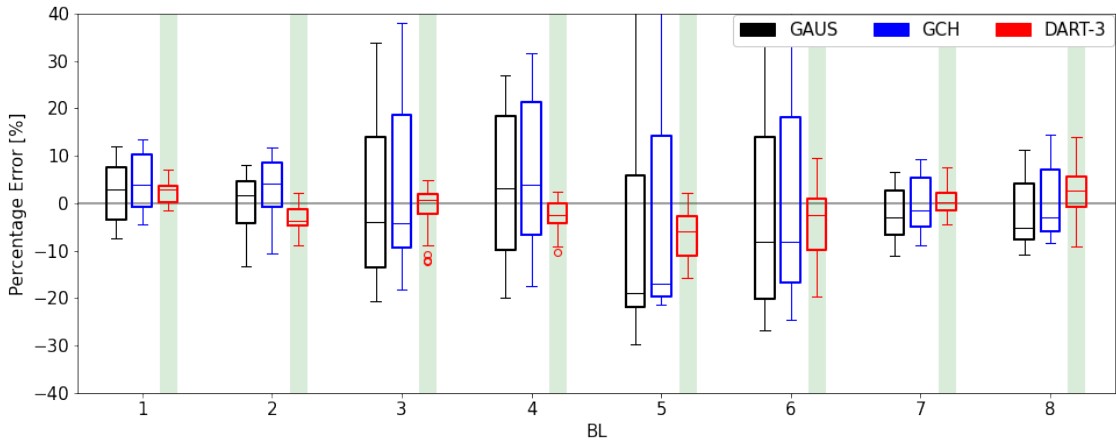

**Figure 9.** Same as Fig. 8, but only for cases with $\beta = 0°$, i.e. without derating the turbine.

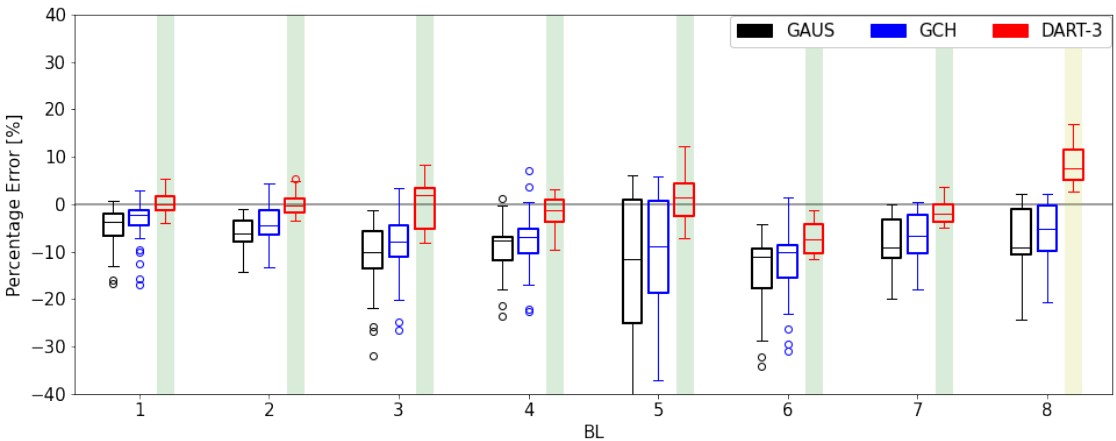

**Figure 10.** Same as Fig. 9, but for partial wake operation, i.e. with a virtual downstream turbine moved 0.5 D to the left.

320   $\phi = +30°$. The wake has a clearly defined curl and a wake center left of the hub. The DART-3 wind field in Figure 11b shows that the wake shape and center position are well presented. The GAUS model (Fig. 11c), however, produces a circular wake shape and a larger wake deflection to the left. The percentage errors indicated in the top of the figure show that DART-3 has a high accuracy for both virtual turbines, but GAUS has large biases due to the misplacement of the wake center. Under stable conditions the wind veer is relatively high, adding a crosswise force pointing towards to right above hub height. This force

325   effectively opposes the lateral thrust force component introduced by yaw misalignment pointing to the left, reducing the deflection of the wake. The opposite is true for negative yaw angles, where wake deflection is enhanced by veer. This asymmetry

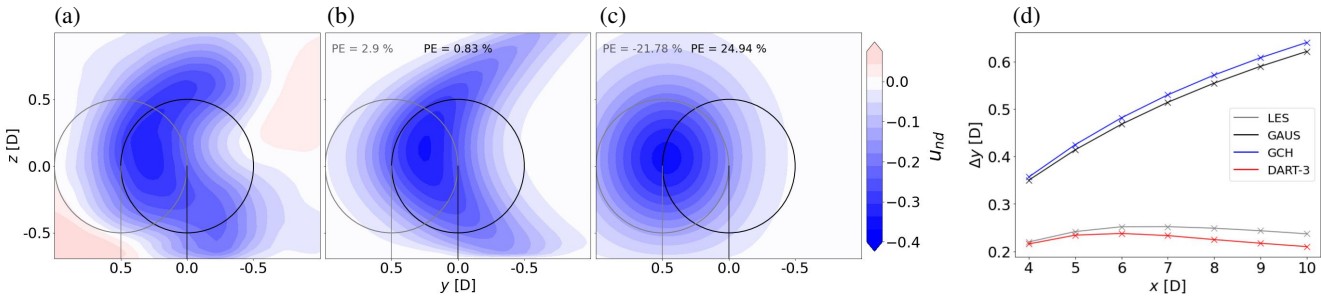

**Figure 11.** Case study of a turbine in a weakly stable boundary layer (BL3, $\phi = +30°$, $\beta = 0°$). Cross section of normalized wake deficit (contours) of the LES (a), DART-3 (b) and GAUS (c) at $x = 6$ D downstream. (d) Wake center trajectory at $4$ D $\leq x \leq 10$ D.

has already been pointed out in Fleming et al. (2015); Vollmer et al. (2016); Sengers et al. (2020). This effect is implicitly included in DART-3, but not in the benchmark models. Figure 11d illustrates that these models show an ever further deflecting wake, whereas DART-3 settles at a smaller lateral displacement close to LES. Not only does this explain the negative bias of the benchmark models in Fig. 10, but also their larger spread observed in Fig. 9. This result strengthens the previous indication that DART-3 is superior under stable stratifications.

## 5  Discussion

### 5.1  Generalizability

Although the results presented in Sect. 4 are encouraging and are believed to show proof of concept, they are not directly generalizable. A data-driven surrogate model is sensitive to the data used for training, and encountering situations that vary greatly from those used for training can result in large errors. This includes very dissimilar atmospheric conditions, as already illustrated by the strongly stable BL5 in Fig. 8, but extends to other locations (e.g. topography, wind farm layout) and turbine type. Generating a numerical database with more atmospheric conditions, tailored to each location and turbine type is not possible due to the high computational expense of these high fidelity models. This limits a large-scale implementation of data-driven surrogate models trained with numerical data. Potentially, field measurements could be used, either in isolation or in combination with numerical data. Wake data could possibly be obtained from long-range lidars (Brugger et al., 2020) or strain measurements from the turbine's blades (Bottasso et al., 2018). Exploration of these possibilities is deemed an important task for future research.

In this exploratory study, the development of DART was limited to the far wake and a two-turbine setup. If desired, further development of the model is needed to include the near wake, which can for instance be done by including the super-Gaussian description (e.g. Shapiro et al., 2019; Blondel and Cathelain, 2020). An extension to wind farm level could be achieved by for instance applying the superposition principle as done in GAUS and GCH, although the accuracy of DART under disturbed inflow needs attention.

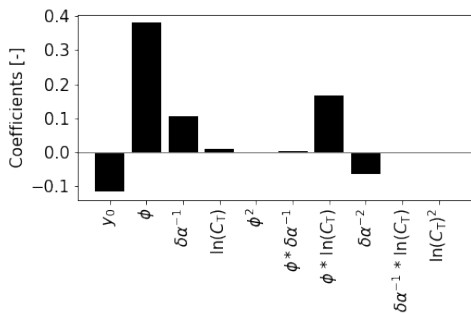

**Figure 12.** Regression coefficients of DART-3 estimating $\mu_y$ at $x = 6$ D using scaled input parameters. Since all input parameters are dimensionless, the corresponding coefficients are also dimensionless. Variable $y_0$ indicates the intercept or systematic offset.

## 5.2 Interpretability

As mentioned in Sect. 1, analytical models such as GAUS and GCH are presumed to be more robust than purely data-driven models. However, when properly trained, the accuracy of DART is expected to be significantly higher than that of analytical models, as it is specifically tailored to certain scenarios. This can easily be understood by looking at the number of fitted or tuned parameters. Since DART includes second-order polynomial and interaction terms, adding more input variables exponentially increases the size of coefficient matrix $\boldsymbol{B}$ (Eq. 1). This means that for DART-3, having only 3 input variables, $\boldsymbol{B}$ contains 10

coefficients, but with 8 input variables DART-8's $\boldsymbol{B}$ already contains 45 coefficients. When comparing this to the 4 tuning parameters of the benchmark models, one can understand why the latter are more robust, but also are expected to have a lower maximum achievable accuracy.

To demonstrate DART's interpretability, Figure 12 illustrates DART-3's fitted regression coefficients for all 10 input parameters for $\mu_y$ at $x = 6$ D. Since the order of magnitude of the input parameters can vary greatly, for this example the input parameters

were scaled between -1 and 1 before regression fitting. Consequently, the fitted coefficients indicate how important each input parameter is in estimating the output variable. For the lateral wake center displacement it can easily be seen that $\phi$ is the dominant parameter, which intuitively makes sense. Other import parameters are the interaction term $\phi * \ln(C_{\mathrm{T}})$ (turbine variable cluster), $y_0$ (intercept), $\delta\alpha^{-1}$ and $\delta\alpha^{-2}$ (atmospheric inflow cluster), while other parameters only slightly affect the wake center displacement.

Alternative to the interpretable Lasso model, more complex black-box models (e.g. neural networks) could be considered as they are expected to have a higher accuracy when abundant data is available. Simpler models are, however, always preferred because they are less prone to overfitting, which is especially true for small sample sizes as used in this study. In addition, a model's inpretability typically diminishes with increasing complexity.

**Table 5.** Model run time [ms] when simulating seven (4 D $\leq x \leq$ 10 D) and one ($x$ = 6 D) downstream distances expressed as mean $\pm$ standard deviation over 40 iterations.

| $x$ [D] | **4-10** | **6** |
|---|---|---|
| **GAUS** | 58$\pm$2 | 19$\pm$1 |
| **GCH** | 88$\pm$2 | 32$\pm$1 |
| **DART-3** | 81$\pm$4 | 13$\pm$3 |

## 5.3 Speed test

A simple evaluation of computational costs has been carried out to ensure that DART is sufficiently computationally efficient. The speed test comprises of producing cross sections downstream of the turbine and therefore excludes the computational resources needed to generate the LES data and to train or tune the models. This test was executed on a laptop running Ubuntu 20.04.1 with eight 1.80GHz Intel i7-8550U CPU's and 8 GB RAM, having a minimum number of processes running in the background. All files containing relevant information, such as inflow variables, were stored locally at the same location. Run

times are given as an average and standard deviation over 40 iterations, representing all simulations with $\beta = 0°$, such that no adjustment of the benchmark's thrust coefficient lookup table is needed. Table 5 shows that when producing results for the whole region considered in this study (4 D $\leq x \leq$ 10 D), the run time of DART is comparable to GCH and slightly higher than GAUS. When simulating only one downstream distance, for instance exactly where a turbine is located, DART performs similarly to GAUS. These results suggest that DART is quick enough to be used for controlling purposes.

## 6 Conclusions

This study explores the potential of a Data-driven wAke steeRing surrogaTe model (DART) that retains a high degree of physical interpretation. After training with large eddy simulation data, a model consisting of only linear equations is able to accurately describe the far wake in terms of trajectory, curl and available power. As input parameters, it uses measurable inflow and turbine variables that are commonly studied in literature. The highest accuracy is obtained when including all available

input variables, but the model's training time becomes very large. When using only three measurable input variables, the surrogate model displays a slight accuracy loss, but the training time is greatly reduced. In a benchmark against the Gaussian and Gaussian-Curl Hybrid models, the data-driven model with three input variables typically shows a significantly higher accuracy. In particular it performs better under derated operating conditions and stable atmospheric stratifications, since it implicitly includes the effect of turbine derating on wake size, as well as the effect of veer on the wake center position. These

results are not directly generalizable to all atmospheric conditions, other locations or new turbine types, which presents a challenge for a large-scale implementation of data-driven surrogate models. The results shown in this study are, however, believed to show proof of concept for physically interpretable data-driven surrogate models for wake steering purposes.

*Code availability.* The Data-driven wAke steeRing surrogaTe model (DART), including a short tutorial, is available for download at: https://github.com/LuukSengers/DART

 **Appendix A: Multi-task Lasso algorithm**

The original Lasso implementation from Tibshirani (1996) seeks to find the coefficients $B$ based on

$$\operatorname*{argmin}_{B} \sum_{n} (y_n - \sum_{p} x_{np} B_p)^2 + \lambda \sum_{p} |B_p|. \tag{A1}$$

in which $n$ is the sample size and $p$ the input parameter. It uses the regularization parameter $\lambda$, leading to sparse coefficients for the coefficient vector $B$. The multi-task setting from Obozinski et al. (2006) extends the Lasso regression to estimate $d$ (distance downstream) outputs simultaneously, penalizing the blocks of coefficients over the tasks. The loss function is therefore extended and finds the coefficient matrix $B$ based on

$$\operatorname*{argmin}_{B} \sum_{d} \sum_{n} (y_{nd} - \sum_{p} x_{np} B_{pd})^2 + \lambda \sum_{p} \sqrt{\sum_{d} (B_{pd})^2}. \tag{A2}$$

In contrast to Eq. (A1), the multi-task Lasso implementation does not only penalize the single coefficients, but also the blocks of coefficients over all tasks represented by the Euclidean norm. Note that if $d = 1$, Eq. (A2) reduces to the standard Lasso estimate of Eq. (A1).

An exemplary result is illustrated in Fig. A1. Whereas the original Lasso model selects a new set of variables for each distance, the multi-task Lasso always takes the same set. This makes physically more sense and leads to fewer variables in total, therefore reducing the risk of overfitting. The model is optimized using the cyclical descent algorithm implemented in Pedregosa et al. (2011).

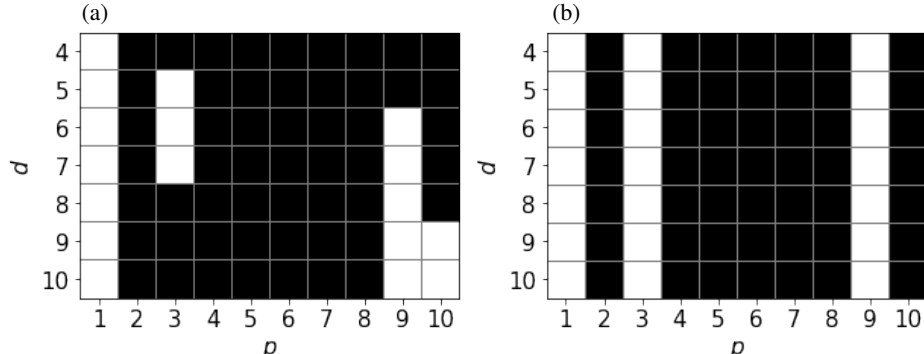

**Figure A1.** Non-zero elements (black) of $B$ for the output variable $\mu_y$ in DART-3 for the original Lasso (a) and multi-task Lasso (b).

*Author contributions.* BS developed the surrogate model, performed the simulations, generated the results and wrote and edited the manuscript. MZ developed the regression model and contributed to writing Sect. 3.3 and Appendix A. PJ tested various tuning strategies for the benchmark models and provided a general understanding of these models. GS provided intensive consultation on the development of the model and the scientific analyses. MK provided general consultation and had a supervisory function. All coauthors reviewed the manuscript.

*Competing interests.* The authors declare that they have no conflict of interest.

*Acknowledgements.* The authors would like to thank Detlev Heinemann, Paul van der Laan, Lukas Vollmer and Andreas Rott for their contributions in discussions about the direction of this research. David Bastine is thanked for providing insights on feature selection. The presented work has been carried out within the national research project "CompactWind II" (FKZ 0325492H) funded by the Federal Ministry for Economic Affairs and Energy (BMWi) on the basis of a decision by the German Bundestag. The work was partially funded by the Ministry for Science and Culture of Lower Saxony through the funding initiative "Niedersächsisches Vorab". M. Zech has been supported by the Deutsche Bundesstiftung Umwelt (grant no. 20020/667-33/2). Computer resources have been provided by the national research project "Heterogener Hochleistungsrechner für windenergierelevante Meteorologie- und Strömungsberechnungen (WIMS-Cluster)" (FKZ 0324005) funded by the Federal Ministry for Economic Affairs and Energy (BMWi) on the basis of a decision by the German Bundestag.

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
