# Peer review of "A physically interpretable statistical wake steering model"

_Wind Energy Science, 2021_

## Referee Comment (RC1)

**Referee report:**

**"A physically interpretable statistical wake steering model"**

**by Balthazar Sengers, Matthias Zech, Pim Jacobs, Gerald Steinfeld, and Martin Kühn**

**1  General comments**

This study presents a data-driven model to model the physics of a wind turbine wake under yawed conditions. The model employs a regression model, with inputs from large eddy simulation (LES) data, resulting in a linear model of the wake behavior. They then evaluate the performance of this model, as well as that of two other wake models, the Gaussian wake model and the Gaussian-Curl Hybrid model, against LES data. All of these models require tuning and/or training, so the same LES data was used for to prepare the models. Then a single case LES case that was left out of the training data was used for evaluation. The main focus of the study is to present a data-driven model that includes more physics and provides a linear model.

This paper provides an interesting new data-driven model, which results in a linear model for the wake behavior of yawed turbines. In particular, the scaling of the input parameters to enable the application of the model to a wide array of atmospheric boundary layer conditions is interesting, as changing conditions have been challenging for data-driven models. The paper presents an interesting approach and I would recommend for publication after minor revisions. Below is a detailed list of comments that should be addressed in a revision of this manuscript:

**Specific comments**

1. In line 66: the authors reference 'default numerical schemes' when describing the LES code. A more detailed description of these should be provided.

2. The caption for Figure 1: the authors give results for 'over the 15 main simulations'. It was unclear which simulations these were, specifically pertaining to their number. In Table 1, eight simulations are listed, so this is a confusing statement.

3. Section 3.4 seems very similar to the description in Section 3.1 and comes off a bit repetitive. Could the differences between them be clarified more?

4. In line 298: the authors mention '...which is due to the 'top head' shape of the wake deficit as a result of temporal averaging.' I'm familiar with the 'top-hat' wake shape but not the 'top-head'. If not a typo, the author should provide a more detailed description of this shape.

5. In line 235: 'To test whether a higher accuracy is achieved when more variables are included, allSWSM uses all (non-transformed) available variables of Table 3 as input.' The authors mention using all the variables rather than three. What (if any) is the time savings in training using only three variables versus all of the variable in Table 3?

6. In line 238: the authors mention that 'since the near wake is more dynamic and therefore needs more parameters to explain its variability'. The authors should provide citations

for this. The authors also mention that the near wake requires more parameters for this description. Are these factors already included in the parameters used in the input parameters or are they additional parameters outside this study? Do the authors have thoughts on what these parameters are? In Figure 7, we only see the results from x/D=4 onwards. Does the allSWSM method improve the near-wake performance at all?

7. In line 247: 'The models are trained or tuned with seven out of the eight BLs (Fig. 1) and tested on the remaining one representing a new inflow condition.' The authors mention the training and tuning of the three models compared in this paper. What is the order of magnitude for how long the training takes for this model? Is there a significant difference between the training of this model and the tuning of the other two wake models it is compared with here?

8. In line 302: 'In this study, large eddy simulation data were used to train the model, the generation of which is computationally expensive.' The authors mention that the model needs to be trained using LES data, which is a limitation. Have they considered whether the model could be trained using some combination of data obtained from an operating wind farm? This would save computational resources and customize the model to that specific wind site. The wake cross-sectional area data could possibly be obtained from strain measurements from the wind turbine blades, such as in Bottasso, Cacciola, and Schreiber, Renewable Energy, Vol 116, Part A, 2018.

9. In line 315-16: 'If desired, further development of the model is needed to include the near wake, which can for instance be done by including the super-Gaussian description introduced by Blondel and Cathelain (2020).' To my knowledge, the super-Gaussian description was also used in Shapiro et al. Energies, 2019; 12, no. 15: 2956.

**Technical comments**

1. In the description of Table 1: "...except for the domain size which is extended in streamwise direction". Typo, missing 'the'

2. In lines 60-61: "A precursor without and a subsequent main simulation with one turbine make up the simulation chain." Sentence is a little confusing. Hard to figure out what the simulation looks like.

3. In line 80: "(Sec. 2.2 and averaged over a line of size 2 d in crosswise direction and a period..." Typo, missing a parentheses ) at the end of Sec. 2.2? Also a missing 'the'.

4. In line 125: "An expression for the wake width as function", typo

5. In line 153: The variable Y is mentioned twice, but X is not mentioned at all

6. In line 168: " IN this section..." typo

7. Figure 6: the second plot in the upper right corner of Figure 6 has no labels and is not mentioned in the caption.

---

## Author Comment (AC1)

**Authors' response to "A physically interpretable statistical wake steering model"**

Balthazar Sengers[1], Matthias Zech[2], Pim Jacobs[1], Gerald Steinfeld[1], Martin Kühn[1]
[1]ForWind, Institute of Physics, Carl von Ossietzky University Oldenburg, Küpkersweg 70, 26129 Oldenburg, Germany
[2]German Aerospace Center (DLR), Institute of Networked Energy Systems, Carl-von-Ossietzky-Str. 15, 26129 Oldenburg, Germany
Correspondence: balthazar.sengers@uni-oldenburg.de

**Response to both reviewers**

The authors appreciate the feedback from both reviewers and believe that the manuscript has been improved after implementing the reviewers' comments.

The authors decided to rename the Statistical Wake Steering Model (SWSM) to the Data-driven wAke steeRing surrogaTe model (DART) . Consequently, the title has been changed to "A physically interpretable data-driven surrogate model for wake steering". Additionally, allSWSM has been renamed DART-8, oSWSM has been renamed DART-3 and cSWSM has been omitted, following comment 7 from Reviewer 2. Lastly, some sentences have been rephrased to minimize self-plagiarism, but these additional changes have not been included in the author's response.

The author's response to each of the reviewers' comments (in black) can be found below (in red), as well as the rephrased sentences or added text (in blue).

**Reviewer 1**

General comments

This study presents a data-driven model to model the physics of a wind turbine wake under yawed conditions. The model employs a regression model, with inputs from large eddy simulation (LES) data, resulting in a linear model of the wake behavior. They then evaluate the performance of this model, as well as that of two other wake models, the Gaussian wake model and the Gaussian-Curl Hybrid model, against LES data. All of these models require tuning and/or training, so the same LES data was used for to prepare the models. Then a single case LES case that was left out of the training data was used for evaluation. The main focus of the study is to present a data-driven model that includes more physics and provides a linear model.

This paper provides an interesting new data-driven model, which results in a linear model for the wake behavior of yawed turbines. In particular, the scaling of the input parameters to enable the application of the model to a wide array of atmospheric boundary layer conditions is interesting, as changing conditions have been challenging for data-driven models. The paper presents an interesting approach and I would recommend for publication after minor revisions. Below is a detailed list of comments that should be addressed in a revision of this manuscript:

Specific comments
1. In line 66: the authors reference 'default numerical schemes' when describing the LES code. A more detailed description of these should be provided.
   The authors agree that the model settings should be elaborated upon. When adding this information, the authors realized that this information would better fit in Sect. 2 instead of Sect. 2.1, since this information holds for both precursor and main simulations. The following text has therefore been moved to Sect. 2.
   "In the boundary layer, the grid on a right-handed Cartesian coordinate system is regularly spaced with $\Delta = 5$ m, while above the boundary layer height the vertical grid size increases with 6 % per cell to save computational costs.  The Coriolis parameter corresponds to $55°$N. Default

numerical schemes are used, the main ones being a third-order Runge-Kutta scheme for time integration, a fifth-order Wicker-Skamarock advection scheme for the momentum equations, Deardorff's 1.5-order turbulence closure parameterization for subgrid-scale turbulence and an iterative multigrid scheme for the horizontal boundary conditions."

2. The caption for Figure 1: the authors give results for 'over the 15 main simulations'. It was unclear which simulations these were, specifically pertaining to their number. In Table 1, eight simulations are listed, so this is a confusing statement.
   The caption has been rephrased.
   "Summary of the most relevant inflow parameters (60 min averages), given as mean (dots) and standard deviation (whiskers) over the 15 main simulations performed in each BL (5 yaw angles times 3 pitch angles)."

   Additionally, one sentence in Sect. 2.2 has been altered for clarification
   "This adds up to a total of 120 main simulations with one turbine, i.e. 15 turbine settings (5 yaw angles times 3 pitch angles) for each of the 8 inflow conditions."

3. Section 3.4 seems very similar to the description in Section 3.1 and comes off a bit repetitive. Could the differences between them be clarified more?
   These sections are indeed very similar, since Sect. 3.4 describes what is basically the reverse of Sect. 3.1. The authors agree an extended clarification might be unnecessary. Section 3.4 has therefore been rewritten.
   "The coefficient matrix $B$ can be used to estimate the key wake steering parameters in Table 2 from inflow variables. This information is used to compose a vertical cross section of the wake deficit using the reverse of the Multiple 1D Gaussian method described in Sect. 3.1. The amplitude of the normalized wake deficit $A$ at each height ($k = 1..K$) can be computed by simply filling out the Gaussian function using $A_z$, $\mu_z$, $\sigma_z$. Similarly, local wake center positions $\mu$ and local wake widths $\sigma$ can be found by filling out their second order polynomials. Additional assumptions outside of the rotor area include that the curl remains continuous (red dashed line in Fig. 4c) and the wake width can be described by an ellipse between tip (upper or lower) and wake edge (top or surface), see red dashed line in Fig. 4d. Finally, a simple 1D Gaussian can be filled out at every vertical level using the information from $A$, $\mu$, $\sigma$, resulting in a two-dimensional grid filled with $u_{nd}$ values (Fig 4e)."

4. In line 298: the authors mention '...which is due to the 'top head' shape of the wake deficit as a result of temporal averaging.' I'm familiar with the 'top-hat' wake shape but not the 'top-head'. If not a typo, the author should provide a more detailed description of this shape.
   This was indeed a typo and has been corrected.

5. In line 235: 'To test whether a higher accuracy is achieved when more variables are included, allSWSM uses all (non-transformed) available variables of Table 3 as input.' The authors mention using all the variables rather than three. What (if any) is the time savings in training using only three variables versus all of the variable in Table 3?
   The authors acknowledge that the impact of different DART configurations on the training time could be interesting to the reader. This comment has been combined with comment 7, and a speed test for training the different models and model configurations has been added. This speed test was added to Sect. 4.1 as part of the results

Table 4. Model training (DART) or tuning (GAUS and GCH) time when training with all 120 simulations and 7 ($4\,D \leq x \leq 10\,D$) down-stream distances. *Iteration* times are expressed as the mean over the first 100 iterations. DART's *Total* is a simple multiplication of *Iteration* and *Combinations*.

| | Iteration | Combinations | Total |
|---|---|---|---|
| | [s] | [-] | [h] |
| **GAUS** | - | - | 7.5 |
| **GCH** | - | - | 8.25 |
| **DART-8** | 148 | 234375 | 9635 |
| **DART-3** | 45 | 900 | 11.25 |

"The order of magnitude of computational costs needed to train the models on a single node is displayed in Table 4. Computational expenses needed to generate the LES database are not considered. The benchmark models tune their parameters in approximately 7.5 h (GAUS) and 8.25 h (GCH). DART's training procedure is split up in different stages. The column *Iteration* refers to the regression fitting to obtain the coefficient matrix $B$ (Sect. 3.3) and the calculation of the absolute percentage error of available power at $4\,D \leq x \leq 10\,D$ (Sect. 3.5). This can be carried out in seconds, in which fewer variables result in faster fitting. The column *Combinations* indicates the number of possible combinations that need to be tested (Sect. 3.5). The *Total* training time is then simply the number of combinations to be tested times the execution time of one iteration. Because of its large number of possible combinations, DART-8's total training time would be over a year on a single node, which is not operationally feasible. To generate the results in Fig. 7, the training process was heavily parallelized. With 900 possible combinations DART-3 can be trained in approximately 11.25 h, which is only slightly more than the benchmark models. As mentioned in Sect. 3.5, the training procedure could be enhanced, but this was considered outside of the scope of the current work. "

6. In line 238: the authors mention that 'since the near wake is more dynamic and therefore needs more parameters to explain its variability'. The authors should provide citations for this. The authors also mention that the near wake requires more parameters for this description. Are these factors already included in the parameters used in the input parameters or are they additional parameters outside this study? Do the authors have thoughts on what these parameters are?
The authors admit that this statement was actually incorrect. Rather than the near wake being physically more dynamic, the authors meant that the range of possible values for the key wake steering parameters, in particular the wake center deficit, is larger.
"The accuracy of DART-8 is higher than that of DART-3, especially closer to the turbine. This is attributed to the stronger wake deficit closer to the turbine, as the wake center deficit $A_z$ exhibits a larger range of possible values closer to the turbine. For instance in the training data at $x = 4\,D$, the range is $-0.6 \leq A_z \leq -0.21$, whereas at $x = 10\,D$ the range is $-0.27 \leq A_z \leq -0.09$ . Estimations with the same relative error therefore bear a larger absolute error closer the turbine. Having access to more information, DART-8 consistently has a smaller relative error estimating $A_z$ than DART-3, which has a larger effect on the available power estimates closer to the turbine."

In Figure 7, we only see the results from x/D=4 onwards. Does the allSWSM method improve the near-wake performance at all?

As discussed in Sect. 3.4.1, DART gives high errors for $x < 4$ D due to the double Gaussian profile of the near wake. For this reason, the near wake is not further considered and DART-8 is only trained for $4$ D $\leq x \leq 10$ D. To fully answer the reviewer's question, DART-8 would need to be retrained including the near wake, but due to the enormous computation expense as discussed in the previous comment, the authors decided not to do this. It is, however, expected that DART-8 would provide more accurate results than DART-3 in the near wake, simply because more accurate estimates of the key wake steering parameter can be made, see discussion above. The double Gaussian, however, is also not resolved with DART-8.

7. In line 247: 'The models are trained or tuned with seven out of the eight BLs (Fig. 1) and tested on the remaining one representing a new inflow condition.' The authors mention the training and tuning of the three models compared in this paper. What is the order of magnitude for how long the training takes for this model? Is there a significant difference between the training of this model and the tuning of the other two wake models it is compared with here?
This comment has been addressed in comment 5.

8. In line 302: 'In this study, large eddy simulation data were used to train the model, the generation of which is computationally expensive.' The authors mention that the model needs to be trained using LES data, which is a limitation. Have they considered whether the model could be trained using some combination of data obtained from an operating wind farm? This would save computational resources and customize the model to that specific wind site. The wake cross-sectional area data could possibly be obtained from strain measurements from the wind turbine blades, such as in Bottasso, Cacciola, and Schreiber, Renewable Energy, Vol 116, Part A, 2018.
The authors were delighted to read this comment, as this is the topic of the authors' ongoing work. The authors agree that the outlook deserves more attention and thank the reviewer for the suggested citation.
"Potentially, field measurements could be used, either in isolation or in combination with numerical data. Wake data could possibly be obtained from long-range lidars (Brugger et al., 2020) or strain measurements from the turbine's blades (Bottasso et al.,2018). Exploration of these possibilities is deemed an important task for future research."

9. In line 315-16: 'If desired, further development of the model is needed to include the near wake, which can for instance be done by including the super-Gaussian description introduced by Blondel and Cathelain (2020).' To my knowledge, the super-Gaussian description was also used in Shapiro et al. Energies, 2019; 12, no. 15: 2956.
The authors thank the reviewer for this suggestion. The reference has been added.

Technical comments
1. In the description of Table 1: "...except for the domain size which is extended in streamwise direction". Typo, missing 'the'
This has been fixed
2. In lines 60-61: "A precursor without and a subsequent main simulation with one turbine make up the simulation chain." Sentence is a little confusing. Hard to figure out what the simulation looks like.
This has been rephrased

"The simulation chain consists of a precursor simulation to generate realistic inflow conditions and a subsequent main simulation that contains one turbine."

3. In line 80: "(Sec. 2.2 and averaged over a line of size 2 d in crosswise direction and a period..." Typo, missing a parentheses ) at the end of Sec. 2.2? Also a missing 'the'.
   This has been fixed

4. In line 125: "An expression for the wake width as function", typo
   This has been fixed

5. In line 153: The variable Y is mentioned twice, but X is not mentioned at all
   This has been fixed

6. In line 168: " IN this section..." typo
   This sentence has been deleted following other comments.

7. Figure 6: the second plot in the upper right corner of Figure 6 has no labels and is not mentioned in the caption.
   The caption has been updated
   "The subplot in the top right zooms in to $4\,D \leq x \leq 10\,D$. Axes labels correspond to those of the main plot."

**Reviewer 2**

Purely data-driven flow/wake models are indeed 'under-published' over literature – but perhaps for good reasons. Although the approach itself is quite interesting, I would call for caution when calling it 'a wake model' as it reads more like a clever surrogate to map inputs to outputs for the given problem. Accordingly, I would suggest a more methodology-focused tone in the article rather than reading too much into the results as they are most likely not generalizable.

The authors agree that the paper should focus more on the methodology. Therefore the presented methodology on LES references cases will be referred to as "an exploratory study" to show "proof of concept" in the revised manuscript. Additionally, the reviewer's suggestion to call it a surrogate model has been adopted. The following changes have been made:

- Title
"A physically interpretable data-driven surrogate model for wake steering"

- Abstract
"This study explores whether a data-driven surrogate model with a high degree of physical interpretation can accurately describe the redirected wake."
"Even though the results are not directly generalizable to all atmospheric conditions, locations or turbine types, the outcome of this study is encouraging."

- Section 1
"The objective of this study is to explore the potential of a Data-driven wAke steeRing surrogaTe model (DART) that retains a high degree of physical interpretation."
"Lastly, the surrogate model's generalizability to all atmospheric conditions, locations and turbine types is discussed."

- Section 2.1
"Due to the large computational expense it was not possible to increase the number of simulations. Although these eight BLs do not capture the great variability of the free field, it is considered sufficient to provide a proof of concept for data-driven models."
- Section 2.2
"It should be noted that smaller step sizes for yaw and pitch angles would be preferred, as these step sizes might be too coarse when utilizing a regression based model (Sect. 3.3). This can lead to deviating estimates when interpolating to values far away from these set points (e.g. for $\varphi = 7.5°$). Increasing the step size would, however, lead to more simulations, which was computationally not feasible and not deemed necessary to show proof of concept." (Also comment 2)

- Section 3
"It should be noted that many different kinds of data-driven models exist. For the purpose of this pilot study, the focus was to develop a simple regression model that performs well on small data sets without the risk of overfitting."

- Section 4.2
"This indicates a limited generalizability of all models and caution is needed when applying them in conditions that differ greatly from those used for training. This will be further discussed in Sect. 5.1."

- Section 5.1
"Although the results presented in Sect. 4 are encouraging and are believed to show proof of concept, they are not directly generalizable. A data-driven surrogate model is sensitive to the data used for

training, and encountering situations that vary greatly from those used for training can result in large errors. This includes very dissimilar atmospheric conditions, as illustrated by the strongly stable BL5 in Fig. 8, but extends to other locations (e.g. topography, wind farm layout) and turbine type. Generating a numerical database with more atmospheric conditions, tailored to each location and turbine type, is not possible due to the high computational expense of these high fidelity models. This limits a large-scale implementation of data-driven surrogate models trained with numerical data."

- Section 6
"These results are not directly generalizable to all atmospheric conditions, other locations or new turbine types, which presents a challenge for a large-scale implementation of data-driven surrogate models. The results shown in this study are, however, believed to show proof of concept for physically interpretable data-driven surrogate models for wake steering purposes."

1. Introduction: The novelty is clear but the motivation for purely data – driven wake modelling is lacking. Since physics – based, data – informed (through parameter fitting) wake modelling is a well – developed area, with several open access toolboxes/implementations as indicated in the article; why would we go for statistical models with even more parameters to fit? Especially if we have limited data?
The authors agree that the motivation for purely data-driven wake modeling deserves more attention.
"Although analytical models are presumably more robust, especially when the data set is small, the maximum achievable accuracy is also limited as it is not feasible to develop one model for all scenarios. Analytical models will not be able to capture features for which equations were not in place, hence constant updates to the model code are necessary (e.g. Abkar et al., 2018; Bastankhah et al., 2022). With the community demanding that wake models include increasingly more complex features (e.g. the wake curl), data-driven models become interesting as they can directly capture these features when enough data is available."

2. Section 2.3 and 3: Every step in Figure 3 should be explained anyway (again, more methodology oriented focus is recommended than too much weight on the results). It reads as if there exist another surface response modelling between the 'LES input variables' and 'input parameters'. Why is it needed?
The authors understand that a more detailed explanation of Figure 3 could be desired. This has therefore been added, including an explanation that each action refers to an upcoming section.
"Figure 3 displays a flowchart of the training and execution (including testing) procedure. The respective sections in which each step is explained, is indicated between parenthesis. The model is trained with the LES data representing reference inflow conditions (BLs) described in Sect. 2. From the wake data, key wake steering variables are deducted by executing the Multiple 1D Gaussian method explained in Sect. 3.1. Additionally, input variables are extracted at 2.5 D upstream, and several operations are performed (Sect. 3.2) to determine the final input parameters. A multi-task Lasso regression (Sect. 3.3) is subsequently performed to generate a coefficient matrix.
This matrix can be used in the execution (testing) of the model to estimate the key wake steering parameters for new inflow conditions. The same operations as in the training procedure are done on the input variables to obtain the input parameters, after which the linear model (Sect. 3.3) is solved to estimate the key wake steering parameters. A reversed version of the Multiple 1DGaussian model can then be executed (Sect. 3.4) to obtain gridded wake data. During model development, this wake estimation can be compared to the original LES data. One can experiment with different input variables and operations to determine what set of input

parameters gives the most accurate solution (Sect. 3.5). This last step has not been included in Fig. 3 to reduce clutter."

15deg steps in yaw control settings are potentially too broad to have a smooth surface for any type of interpolations – its sensitivity and potential effects should be discussed.
The choice for yaw control in steps of 15 degrees was purely to save computational costs, but this or its implications were indeed not mentioned. The following has now been added to Section 2.2.
"It should be noted that smaller step sizes for yaw and pitch angles would be preferred, as these step sizes might be too coarse when utilizing a regression based model (Sect. 3.3). This can lead to deviating estimates when interpolating to values far away from these set points (e.g. for $\varphi=$ 7.5°). Increasing the step size would, however, lead to more simulations, which was computationally not feasible and not deemed necessary to show proof of concept."

3. Section 3.2: "Although these input parameters might all have their own isolated effect on the wake propagation, they are heavily correlated in LES as shown in Fig. 5… Because of the highly correlated clusters, it is hypothesized that one is able to achieve reasonable accuracy in estimating key wake steering parameters with the regression model as long as each cluster is represented." → High correlation between input and output features are indeed desirable for any regression problem but why a high-correlation among the input variables would help with better accuracy?
The high correlation between input variables does not increase the accuracy of the model. Rather, their high correlation makes them interchangable, as they contain much of the same information.
"A high correlation between variables indicates that they contain much of the same information. Providing the same information multiple times is futile as it will not improve the accuracy. For this reason, it is hypothesized that reasonable accuracy can be achieved with the regression model as long as one variable from each input cluster is included. This would reduce the number of model parameters and would give the user freedom to choose parameters based on preference and availability. However, since the input variables are not perfectly correlated, the information they contain is slightly different and including both variables can increase the model's accuracy. For this reason, two versions of the surrogate model having a different number of variables are experimented with, see Sect. 3.5. Although a high correlation between input variables is usually undesirable in regression problems due to multicollinearity, Sect. 3.3 explains that this is not an issue due to the regression model used in this study."

How these heavily correlated effects are to be isolated so that the results are interpretable?
Lasso provides a low-rank model by shrinking irrelevant relationships to zero. In other words, irrelevant parameters will not be considered by setting its coefficient to zero. Out of the set of possible relationships, Lasso aims to select those parameters which are necessary to explain the output variable. This can be regarded as isolating relevant relationships from irrelevant ones. This is also the reason why multicollinearity is not an issue for Lasso, as it simply selects one of the highly correlated input variables and regards the others as irrelevant. The following is added to Sect. 3.3:
"This guarantees a shrinkage of the number of variables through a regularization parameter found by cross-validation. Relevant input parameters are isolated from irrelevant parameters by multiplying the latter with a coefficient of zero, effectively eliminating them from Eq. 1. Multicollinearity is therefore not an issue in Lasso, contrary to Ordinary Least Squares, as typically only one parameter is chosen from a set of highly correlated input parameters. This

reduces the number of input parameters, increasing the interpretability of the model."

How do you see the distribution of the parameters and their sensitivity in line with this high correlation?
The authors are not sure they understand this comment correctly. The authors assume that with "the distribution of the parameters" the reviewer means the distribution of parameters within the whole range of parameters than can affect wake characteristics, or in other words whether the studied variables could explain the observed variability. The authors are of the opinion that a simplified model like this would never be able to fully explain the variability that is observed in reality or high fidelity simulations. The parameters chosen here are, however, studied frequently in literature and are therefore deemed most important. Their high correlation is not expected to be a problem, as different parameters could potential affect the output variables in a similar way. However, the authors acknowledge that if the correlation would be lower in reality than in the numerical environment used here, information would be lost by eliminating parameters and the quality of the output would be sensitive to what parameters are chosen.
Since the authors are unsure whether this is what the reviewer meant, no additional explanation has been added to the revised manuscript. The authors would like to ask the reviewer for clarification if still deemed important after all revisions.

How these input features are correlated to the outputs in Table 2 should be the actual motivation to define the input parameters (together with their availability) but is not discussed at all…
The authors agree that correlating input and output variables would be a viable feature selection procedure. However, since a sophisticated feature selection procedure was deemed outside of the scope of the current work, the authors chose to simply test all possible combinations. The combination of input parameters that shows the smallest absolute percentage error of available power is then chosen. This is described in more detail in the answers to comment 5, as well as comment 5 from reviewer 1.
"It should be noted that other feature selection procedures could be considered to reduce the computational expense needed for training, but enhancing the training procedure was considered outside of the scope of the current work."

4. Section 3.4: Figure is frequently discussed here so it might help the reader to move it further down in the article, closer to Section 3.4.
The figure number in this comment is missing, but the authors assume the reviewer points at Figure 4. This figure is also frequently discussed in Section 3.1, hence its position. As pointed out by reviewer 1, Sections 3.1 and 3.4 were very repetitive, and changes have been made accordingly.

5. Section 3.5: Firstly, the subsection heading should be re-worded to something like "optimum model architecture" as optimization has many applications in the model fit and flow control discussed in the article. Secondly, the subsection generates more questions than answers – does it mean different set of input parameters might be used for different clusters? So there might be several, case-specific (or BL-specific) oSWSM and/or cSWSM?
The authors agree that the current heading is unclear and has been replaced with "Feature selection". The section has been rewritten to explain how the optimum combination of input parameters is determined, as well as to clarify the differences between DART-8 and DART-3.
"Numerous combinations of input parameters are possible. This includes choosing from the variables presented in Table 3, as well as which of the five transformations proposed in Sect. 3.2 to use. In order to find the most accurate solution, all combinations are tested. The combination

that provides the minimum absolute percentage error of available power over all training data, i.e. all considered simulations and downstream distances (4 D ≤ $x$ ≤ 10 D), is sought. When using all eight variables presented in Table 3, denoted DART-8, only the variable transformations need to be decided. The number of possible combinations is proportional to the number of transformation to the power of the number of variables. All five transformations are tested on all variables except for $\varphi$, for which the logarithmic and square root transformations have been omitted because negative values occur. This results in a total of $3^1 * 5^7 = 234375$ possible combinations. Not only is using all variables computationally expensive, as will be discussed in Sect. 4.1, operationally it is also unlikely that all variables are routinely obtained due to high costs. As hypothesized in Sect. 3.2, using one variable from each input cluster is already expected to produce accurate results. To test this, a version of DART with only three variables, denoted DART-3, is considered. Allowing each variable to be chosen and transformed, the total number of possible combinations is a multiplication of the possible combinations of each input cluster. In total, $(1 * 3) * (4 * 5) * (3 * 5) = 900$ possible combinations are tested to find the optimal set of input parameters."

6. Section 4.1: "...The high correlation between variables is not an issue due to the use of the regression method described in Sect. 3.3" → not true as stated earlier. It is an issue at least to isolate the effects of the change in input parameters, hence an issue for the interpretability.
This statement in the manuscript simply refers to the fact that the Lasso algorithm is able to deal with highly correlated input variables, contrary to for instance ordinary least square. Additional explanation has been given in comment 3.
This particular sentence has been removed from Sect. 4.1 after restructuring. All information on high correlation between input variables is now given in Sect. 3.3.

7. Section 4.1: Given the results in Figure 7, I suggest to omit oSWSM and cSWSM approaches to sharpen the focus of the article and avoid potential use of case-specific models (within already limited parameter space defined in LES). If agreed, all results should be updated with allSWSM.
The authors agree that allSWSM did not get enough attention and the reason to omit it in the rest of results was missing. The main reason why allSWSM is not used in all results is on the one hand its large computational expense (now added to Sect. 4.1 following comment 5 from reviewer 1), as well as the impracticality to measure all these variables in the field.
"Even though Fig. 7 shows a small accuracy gain of DART-8 over DART-3, the computational costs to train DART-8 are much larger and measuring all these variables in the free field is impractical. For these reasons, it is decided to only consider DART-3 in the remainder of this study."
SWSM trained with only three variables was included to demonstrate that one does not need all these variables to generate decent results. Both oSWSM and cSWSM were included to show that different combination of variables can give very comparable results. However, for the authors this is perhaps the least impressive outcome of this study. To sharpen the paper's scope, as the reviewer suggested, it has been decided to omit the results from cSWSM. For brevity, all related changes have not been copied into the author's response. For clarity, allSWSM has been renamed DART-8 and oSWSM has been renamed DART-3.

8. Section 4.2: "… Arguably, this is currently not a major disadvantage of the benchmark models as turbines tend to operate without derating (β = 0º)." → should be omitted as turbines do operate under intentional derating with β > 0º) more often than intentional misalignment for wake steering. Therefore, it is an important achievement to be able capture this.

This sentence has been deleted. The remainder has been slightly rephrased to ensure readability.
"The inclusion of this effect is a notable improvement of DART that is important for control strategies such as axial induction control (e.g. Corten and Schaak, 2003; van der Hoek et al., 2019)."

9. Section 4.2 and 4.3: The main advantage of using data-driven approaches is the reduction of uncertainties. The reduction in the variability of the results should be mentioned/discussed at least once throughout Figures 7, 8 and 9.
The reduction in the variability of the results has been discussed a few times in the original manuscript (line number indicating those in the original manuscript):
- line 243-244: "Most striking is the variability of the benchmark models that is an order of magnitude larger than that of oSWSM. The reason for this will be systematically evaluated in Sect. 4.2 and 4.3."
- line 257-259: "The systematic biases (indicated by the medians) are similar for all models in the order of a few percent, but the variability is greatly reduced in SWSM. The main reason for this is that the benchmark models do not include a pitch angle parameter β."
- line 277-280: "In (weakly) stable boundary layers (BLs 3 to 6) GAUS and GCH still show a large variability and occasionally a large systematic bias, which is not true for SWSM. These results suggest that SWSM outperforms the benchmark models especially under stable stratifications, those conditions where wake steering is deemed most effective."
- line 295-298: "Figure 12d illustrates that these models show an ever further deflecting wake, whereas the SWSM settles at a smaller lateral displacement corresponding to LES. This does not only explain the negative bias of the benchmark models in Fig. 11, but also their larger spread observed in Fig. 10. This result strengthens the previous indication that SWSM is superior under stable stratifications."

All these sentences, albeit slightly rephrased, are still present in the revised manuscript.
In case the authors misunderstood the reviewer's comment, the authors would like to ask the reviewer for clarification.

10. Discussion & Conclusion: A stronger discussion on interpretrability is required here. Great that the computational cost is presented but the number of parameters that is in SWSM as well as the other two benchmark models should also be mentioned.
The authors agree that a discussion on number of parameters is missing. Information has therefore been added.
"As mentioned in Sect. 1, analytical models such as GAUS and GCH are presumed to be more robust, but the maximum achievable accuracy is also expected to be limited. This can easily be understood by looking at the number of fitted or tuned parameters. Since DART includes second-order polynomial and interaction terms, adding more input variables exponentially increases the size of coefficient matrix $\boldsymbol{B}$ (Eq. 1). This means that for DART-3, having only 3 input variables, $\boldsymbol{B}$ contains 10 coefficients, but with 8 input variables DART-8's $\boldsymbol{B}$ already contains 45 coefficients. When comparing this to the 4 tuning parameters of the benchmark models, one can understand why the latter are more robust, but also are expected to have a lower maximum achievable accuracy."

It would also be nice to discuss how explainable the whole procedure is, given all the collective data manipulations, normalizations etc. compared to a black box model.
The authors understand that it is difficult for the reader to see how transparent the model still is. The model's interpretability is therefore discussed using the following example.

Figure 12: Regression coefficients of DART-3 estimating $\mu_y$ at $x = 6$ D using scaled input parameters. Since all input parameters are dimensionless, the corresponding coefficients are also dimensionless. Variable $y_0$ indicates the intercept or systematic offset.

[Figure]

 "To demonstrate DART's interpretability, Figure 12 illustrates DART-3's fitted regression coefficients for all 10 input parameters for $\mu_y$ at $x= 6$ D. Since the order of magnitude of the input parameters can vary greatly, for this example the input parameters were scaled between -1 and 1 before regression fitting. Consequently, the fitted coefficients indicate how important each input parameter is in estimating the output variable. For the lateral wake center displacement it can easily be seen that $\varphi$ is the dominant parameter, which intuitively makes sense. Other import parameters are the interaction term $\varphi * \ln(C_T)$ (turbine variable cluster), $y_0$ (intercept) and $\delta\alpha^{-1}$ and $\delta\alpha^{-2}$ (atmospheric inflow cluster), while other parameters only slightly affect the wake center displacement."

Additionally, do you see a potential trade-off between interpretablity and accuracy, i.e. do you think you might have got a better performance out of a black-box model?
Simpler models are always preferred according to Occam's razor. With (infinitely) large data sets, more complex models (e.g. tree-based models or neural networks) are expected to have higher accuracy than simpler models when applied on the same data set. However, in this study the data set is limited and only present a subset of all plausible states. In this case, a simpler model is preferred as it less prone to overfitting and therefore generalizes better to unseen (new) conditions.
"Alternative to the interpretable Lasso model, more complex black-box models (e.g. neural networks) could be considered as they are expected to have a higher accuracy when abundant data is available. Simpler models are, however, always preferred because they are less prone to overfitting, which is especially true for small sample sizes as used in this study. In addition, a model's inpretability typically diminishes with increasing complexity."